# Adaptive Aging Safety of Guidance Marks in Rail Transit Connection Systems Based on Eye Movement Data

**DOI:** 10.3390/ijerph19020725

**Published:** 2022-01-10

**Authors:** Yong Fang, Wenli Zhang, Hua Hu, Jiayi Zhou, Dianliang Xiao, Shaojie Li

**Affiliations:** 1School of Urban Rail Transportation, Shanghai University of Engineering Science, Shanghai 201620, China; fangyong87@tongji.edu.cn (Y.F.); zhangwenli95@yeah.net (W.Z.); jyzhou0034@163.com (J.Z.); 2Transportation Safety Research Center, China Academy of Transportation Sciences, Beijing 100029, China; sean8205@163.com; 3Shanghai Municipal Engineering Design and Research Institute (Group) Co., Ltd., Shanghai 200092, China; lishaojie@smedi.com

**Keywords:** eye movement data, rail transit, connection system, safety of guidance marks, adaptive aging safety design

## Abstract

The aim of this study was to meet the visual cognition needs of the elderly population for the guidance marks and safety guidance marks of the rail transit connection system. Based on the visual characteristics of the elderly population, this paper firstly determined the visual field and sight range of the marks of the elderly population from three aspects—visual angle, visual distance, and height of the elderly population—and constructed the visual recognition space of the elderly population. Then, from the perspective of the setting position, the setting height, and the deflection angle, an adaptive aging safety design method for the guidance marks in the rail transit connection system is proposed. Then, based on the eye movement data of fixation duration, initial fixation duration, and the number of visits, a visual behavior index model is constructed to iteratively optimize the adaptive aging safety design of guidance marks in a rail transit connection system. A radar map is used to calculate the comprehensive index of visual behavior to determine the optimal scheme. Finally, taking the traffic connection system of Shanghai Songjiang University Town Station as an example, the eye movement data of 37 participants were collected, according to the principle that each connection path should only be taken once per person; the above method was used to design 7 connection path guidance marks for an adaptive aging safety design. The results showed that the comprehensive index of visual behavior of different paths had different degrees of improvement of up to 14.00%, which verified the effectiveness of the design method. The research results have certain theoretical significance and application value for the adaptive aging safety design and retrofit of guidance marks of rail transit connection systems.

## 1. Introduction

As an important part of the transportation system, rail transit connection systems provide the connection of various transportation modes, such as buses, taxis, non-motor vehicles, and trams. The connection paths are numerous and complicated, so the design of transportation guidance marks is important. At present, there are many problems, such as unreasonable location settings, inappropriate identification angles, and low levels of passenger flow guidance services [1,2,3]. The design and setting of guidance marks are affected by visual needs, and the visual needs of different groups of people are different, especially for the elderly population. With the progression of age, the visual function of the elderly population gradually declines [4,5], and the number of traffic accidents among the elderly population often accounts for a large proportion every year. Therefore, in the context of aging population, the safety of the elderly population cannot be ignored, and the safety design of guidance marks of the rail transit connection system is very important.

The elderly population have the characteristics of slow movement and unresponsiveness when observing directions and choosing paths. Improper setting of guidance marks can easily cause the elderly to trip and bump into obstacles, mainly due to the decline of vision and visual recognition ability in the elderly. Therefore, it is very important to study the visual characteristics of the elderly and the time of visual recognition for guidance marks to prevent accidents in the elderly. The current research on the design of guidance marks mainly focuses on ordinary pedestrians, without considering the differences in human factors. Considering this, it is necessary to design the guidance marks in rail transit connection system for adaptive aging safety. It is a design concept is that based on the physiological characteristics of the elderly population and can meet their safety needs.

Through the analysis of the visual characteristics of the elderly population and construction of the identification visual range angle, the spatial system of visual recognition of the elderly population was proposed to determine the visual range of the elderly population. Based on the eye movement data, the visual behavior index model was proposed to judge the visual recognition time of the elderly population. Considering human factors, the adaptive aging safety design of the guidance marks in the rail transit connection system is conducive to ensuring the safety of the elderly population and improving the level of travel services available to them.

## 2. Research Overview

Guidance marks design is an important part of pedestrian flow design. In the rail transit connection system, passengers can search and choose a path according to the information provided by the guidance marks. Improper design of guidance marks directly affects the path choice of pedestrians and the time pedestrians take to reach their destination. The current research on pedestrian guidance marks design is mainly reflected in four aspects, as follows: personal wayfinding capabilities, guidance auxiliary equipment, guidance marks system evaluation, and guidance marks design for particular places and for particular people.

Lawton studied the individual wayfinding ability, based on the individual characteristics of passengers [6]. Pazzaglia, Kato, and Prestopnik considered the characteristics of human factors and, on the basis of these, studied the impact of mark design on pedestrians and path choice decision making from the aspects of spatial perception [7], direction sense [8], and the familiarity of marks [9]. Mtinzer started from a variety of environments to study the main auxiliary facilities for passenger flow to the destination [10]. Li summarized the problems and information content of the static mark of the hub, and proposed setting heights of the guidance marks and viewing distances of the guidance marks [11], based on walking speeds. Lin proposed two goals—“the direction mark should meet the optimal level of guidance in space” and “the shortest distance of the guidance”—and built a model to optimize it, using the non-dominated sorting genetic algorithm with business strategy [12], providing a new idea for the design of pedestrian guidance marks. Robins and Holmes et al. [13] discovered the diversity of pedestrian identification methods for information credibility, and designed comparative experiments to explore the influence of pedestrians’ visual preferences on information credibility. Bosch and Gharaveis focused on the cognitive and visual impairment of the aging population, summarized the current dilemmas faced by this group of people, and listed a small number of solutions based on technological means or design strategies [14].

In terms of the locations for guidance marks designs, the existing research mostly focuses on rail transit stations [15], airport terminals [16], shopping malls [17,18], universities [19], and other places. Additionally, there has not been much attention paid to the rail transit connection system. In the most representative research, Li et al. explored the short-term memory ability and short-term memory differences of passengers for different information of subway marks. Based on the STM theory, they outlined the results of the short-term memory characteristics of subway marks, but they did not provide a detailed solution [20]. In their research, Lee et al. found that the Seoul subway station caused visual fatigue to passengers due to excessive guidance marks [21]. Fewings studied the problem of wayfinding in an airport terminal and discussed how passengers use the guidance marks as clues to find the way, but the study did not give a practical solution on how to improve the wayfinding process in the terminal [22]. Based on the current chaotic situation of guidance marks on university campuses, Kim proposed a plan to replace traditional marks with digital marks [23]. When Jonathan was studying the guidance marks at the exit of the highway tunnel, based on the driving simulator experiment, he simulated the optimization plan for the height and angle of the guidance marks at the exit of the tunnel [24].

In summary, existing research subjects are mostly ordinary passengers, and extant studies do not consider the differences in human factors; additionally, the safety issues of guidance marks are often ignored. Therefore, this paper takes the elderly population as the research subject, in order to formulate an adaptive aging safety design for guidance marks in the rail transit connection system.

## 3. Methods

### 3.1. Analysis of Visual Characteristics and Construction of Visual Cognition Space System in the Elderly Population

People’s visual characteristics are influenced by visual range, which is determined by people’s visual field and visual distance [25]. The visual field is often represented by an angle and the visual range of the mark is related to the height of the person [26]. Based on the visual characteristics of the elderly population and mark recognition logic, this paper constructs a visual recognition space system for the elderly population, which can be understood as a three-dimensional space, within which, the human eye can capture the mark and clearly identify the information on the mark. Therefore, the visual recognition space system is composed of three parts: the visual field, the visual range of the mark, and the visual recognition information of the mark. The visual field of the elderly population is determined from the three aspects: the visual angle, the visual distance, and the height of the elderly population. The visual range of the mark and design parameters, such as the layout elements of the mark itself, allows identification of the visual recognition information. The visual range can be determined from the deflection angle, the height, and the position of the mark. These factors are necessary in constructing the visual recognition space system of the elderly population. Among the above parameters, the visual angle is mainly based on medical survey statistics and the regulations of the National Fire Protection Association (NFPA). Visual distance refers to the distance between the person and the mark when the information of the mark can be clearly identified, mainly considering the deflection angle and the font width of the mark. The height of the elderly population is based on the average height of the elderly population aged 60~69. The specific design steps are outlined below.

Design steps:

Step 1: Mark setting position. According to the medical survey statistics and the regulations of the National Fire Protection Association (NFPA) [27], the minimum visual angle of people is 0.29°, the longitudinal visual field range is between 50° and 70°, and the maximum transverse visual field range is about 62°. In this paper, the visual field range of the elderly population is the minimum of the normal visual field range of ordinary people, that is, the longitudinal visual field is 50° up and down, and the transverse visual field is around 50°. From Figure 1, point A is the human eye, ∠BAD is the longitudinal visual field, ∠CAE is the transverse visual field, both sides of the surface have o as the center are hemispheres and cones, respectively, and the cones and hemispheres together constitute the visual field of the elderly population. In this range, the human eye can clearly capture the mark, and the mark setting location is effective when the mark is in this range.

Step 2: Mark setting height. The height of the mark setting is mainly based on the average height of the elderly population. According to the research performed in [28], the average height of the elderly population is 160±0.07 cm; therefore, the average height of 160 cm was adopted in this study. Therefore, this paper took the bottom edge height of the mark as the setting height of the mark, dictating that, if the actual height of the mark is in the visual field of the elderly person, then the mark height is considered effective.

Step 3: Mark deflection angle.

A: If the visible range of the mark is defined as within this range, the human eye can clearly identify the text information on the mark. The visual range model of the mark is constructed as shown in Figure 2a. The visual range of the mark is calculated as shown in Figure 2b. Therefore, if a guidance mark is in the visual field of a person, and the human eye is within the visual range of the mark—that is, the human eye can capture the mark and clearly identify the text information on the mark—then the mark deflection angle is considered to be effective.

B: Visual range model for mark.

As shown in Figure 2a, a two-dimensional coordinate system is established. The coordinates of points C, A, and B are C (0, 0), A (−*b*, 0), B (*b*, 0), respectively, and let E (*x*, *y*). From the line segment expression and trigonometric identities, we can obtain the following:(1)$$\mathrm{Line}\;\mathrm{AE}:\;m_{1}=\frac{y}{x+b}$$
(2)$$\mathrm{Line}\;\mathrm{BE}:\;m_{2}=\frac{y}{x-b}$$
(3)$$\tan(\varphi )=(m_{1}-m_{2})/(1+m_{1}m_{2})$$

According to Formulas (1)–(3), we can obtain the following:(4)$${\left(\frac{b}{\sin(\varphi )}\right)}^{2}=x^{2}+{\left(y-\frac{b}{\tan(\varphi )}\right)}^{2}$$

According to Formula (4), the figure formed by point E is a circle with (0, tan(*φ*)) as the center and $\frac{b}{\sin(\varphi )}$ as the radius. Therefore, as shown in Figure 2b, the visible range of the mark is a sphere.

Step 4: Mark layout design.

The mark layout elements are designed in accordance with the corresponding specifications, including the type, shape, size, color, graphic, and text. This paper uses eye movement indicators to indirectly characterize the impact of the mark information on the visual recognition of the elderly population.

### 3.2. Construction of Visual Behavior Index Model Based on Eye Movement Data

Under the visual condition, the mark can be recognized, and the effectiveness of the mark design can be evaluated through the changes of the eye movement data after the mark is recognized. Therefore, this paper proposes a visual behavior index, based on eye movement data, which refers to the proportion of the eye movement index value of a certain guidance mark in the rail transit connection system, in the sum of the eye movement index values of all the marks in the entire system. In this paper, a visual behavior index model was constructed by the visual behavior index of fixation duration, the visual behavior index of initial fixation duration, the visual behavior index of visits, and a comprehensive visual behavior index.

#### 3.2.1. Visual Behavior Index of Fixation Duration

In the process of rail transit connection, the fixation time refers to the duration of pedestrians use of a guidance mark—that is, the duration of time spent at the mark. Suppose the fixation time is *T*, then the visual behavior index of a path can be obtained as follows:(5)$$DVBI^{p}=\frac{VT^{p}}{TT^{p}}$$

Among them:(6)$$\{\begin{array}{c} VT^{p}=\sum_{i\in v}\sum_{k\in v}Z_{ik}^{p}T_{ik}^{p}\\ TT^{p}=\sum_{i\in v}\sum_{k\in v}Z_{ik}^{p}(\sum_{i\in u_{ik}}N_{t}T_{t}) \end{array}$$

$DVBI^{p}$ represents the fixation duration and visual behavior index of the path, p; $VT^{p}$ represents the time that the path, p, stays on the guidance mark pointing to the destination; $TT^{p}$ represents the total fixation time of all guidance marks in the path, p; $Z_{ik}^{p}$ represents the 0–1 variable, which passes through the starting and ending path, p, of arc connection at two arbitrary nodes, *i* and *k*—if you look at the guidance mark information from *i* to *k*, its value is 1, otherwise, it is 0; $T_{ik}^{p}$ indicates the duration spent looking at the guidance marks from *i* to *k*; $N_{t}$ represents the *t*-th guidance mark from *i* to *k* arc segments; $T_{t}$ is the fixation duration of the *t*-th guidance mark of arc segments *i* to *k*; $u_{ik}$ represents the set of mark nodes included in connecting *i* to *k* in addition to *k*.

#### 3.2.2. Visual Behavior Index of Initial Fixation Duration

The first fixation duration in different connection modes represents the length of time taken for pedestrians to fixate on the first guidance mark in the connection process. Assuming that the length of the initial fixation is $T_{0}$, since pedestrians go to the subway station according to the designed path, different paths have different times of seeing the guidance mark for the first time; therefore, only the initial fixation duration is considered to obtain the visual behavior index of the initial fixation duration for a certain path, as follows:(7)$$FVBI^{p}=1-\frac{VT_{0}^{P}}{TT^{p}}$$

$FVBI^{p}$ represents the visual behavior index of the initial fixation duration of the path, p; $VT_{0}^{p}$ represents the time length of the fixation on the path, p, to the first guidance mark pointing to the destination.

#### 3.2.3. Visual Behavior Index Visits

The number of visits in the process of rail transit connection represents the frequency of pedestrians looking at a certain guidance mark. That is, the visual behavior index of access times of a path is obtained as follows:(8)$$NVBI^{p}=\frac{VN^{p}}{TN^{p}}$$

Among them:(9)$$TN=\sum_{i=1}^{n}N_{i}$$

$NVBI^{p}$ represents the visual behavior index of the number of visits of the path, p; $VN^{p}$ represents the number of visits to the destination guidance marks in the path, p; $TN^{p}$ represents the total number of visits of all the guidance marks in the path, p.

#### 3.2.4. Visual Behavior Comprehensive Index

(1)Radar map drawing: Through Excel, the visual behavior index of fixation duration, the visual behavior index of initial fixation duration, and the visual behavior index of visits were projected on the corresponding index number line and connected into lines to draw a radar chart.(2)Characteristic quantity calculation: The average area and perimeter of the radar graph were used to construct a comprehensive index of visual behavior.


(10)
$$\{\begin{array}{cc} {\bar{S}}_{p}=\frac{\sum_{l=1}^{n-1}\sum_{l<j}^{n}\frac{1}{2}R_{l}R_{j}\sin\theta_{lj}}{c_{n}^{2}} & \;\mathrm{Areavector}\\ {\bar{V}}_{p}=\frac{{\bar{S}}_{p}}{\pi {\left(\frac{{\bar{L}}_{p}}{2\pi }\right)}^{2}} & \;\mathrm{Perimeter} \end{array}$$


Among them:(11)$${\bar{L}}_{p}=\frac{\sum_{l=1}^{n-1}\sum_{l<j}^{n}\sqrt{R_{l}^{2}+R_{j}^{2}-2R_{l}R_{j}\cos\theta_{lj}}}{c_{n}^{2}}$$

In the formula:

p—proposed evaluation path;

${\bar{S}}_{p}$—average area of radar chart;

${\bar{V}}_{p}$—perimeter vector;

${\bar{L}}_{p}$—perimeter of the radar chart;

n—index number;

R—side length of the indicator axis;

l,j—the i-th and j-th index;

θ—the angle between two sides.

The comprehensive index of the visual behavior of each path can be calculated as follows:(12)$$Y_{p}=\sqrt{{\bar{S}}_{p}\times {\bar{V}}_{p}}$$

### 3.3. Design Method

It was found that the rail transit connection system guidance marks have validity problems in terms of setting position, setting height, and deflection angle, by consulting related documents and investigating the rail transit connection system guidance marks. Combined with the visual needs of the elderly population, this paper puts forward a visual behavior index, based on eye movement data, aiming to improve the passenger flow connection efficiency. This carries out the adaptive aging design from the problems existing in setting position, setting height, and deflection angle, and uses the visual behavior comprehensive index to evaluate the effectiveness of the optimized design. The design process is shown in Figure 3.

## 4. Results

### 4.1. Analysis of Current Situation

Taking Songjiang University Town Station of Shanghai Metro Line 9 as an example, the transportation modes connected with rail transit mainly include buses, taxis, non-motor vehicles, and tram cars, with a total of 7 connection paths. The current situation of rail transit connection in Songjiang University Town Station is shown in Figure 4.

#### 4.1.1. Mark Design for Adaptive Aging

Based on the visual recognition space system of the elderly population, the guidance marks in the connection system of Songjiang University Station are optimized. Taking path 3 as an example, the guidance marks for non-motor vehicles are designed for adaptive aging. Since the non-motor vehicle area is blocked by trees and other obstacles, we propose that the marks are moved to an obstacle-free location, and that it is ensured that the guidance marks are in the visual field of the elderly population who are parking non-motor vehicles. At the same time, the deflection angle should ensure that the non-motor vehicle parking area is within the visual range of the guidance marks. The adjustment of the design index is shown in Figure 5.

As shown in Figure 5, the first adaptive aging design result of the guidance mark on path 3 is 2.73 m in height, 15.81 m away from pedestrians, and has a 37° deflection angle.

#### 4.1.2. Iterative Optimization Design

Considering the eye movement index, an eye movement experiment was carried out at Songjiang University Town Station. The experimental instrument adopted Tobii Pro fusion, a new generation of high-performance portable eye trackers with corneal reflex and binocular stereo dark pupil tracking technology, with a sampling frequency at 100 Hz. The subjects were 37 elderly people, with an average age of 63, who took rail transit 2–5 times a week. The subjects had good vision, no color blindness or color weakness, and no physical defects affecting their visual faculties. The experiment mainly studied the changes in the eye movement data of subjects by changing the setting height, the setting position, and the deflection angle of the guidance marks in the connection path. During the experiment, subjects were asked to wear the eye tracker. In order to avoid the impact of repeated experiments on the objectivity of the data, we followed the principle that each connection route should only be taken once per person. The connection experiment was carried out at Songjiang University Town Station and the eye movement data was collected as shown in Table 1.

After the eye movement experiment, the eye movement data was imported into Tobii Pro Lab software for processing. Tobii Pro Lab software is an eye tracking software used for experimental research which is manufactured by Tobii headquarters in the greater Stockholm area in Sweden. Through the analysis of heat map and radar graph indicators, the most frequent patterns of the subjects were understood. The hotspot map reflected information including, for example, the location of the focus and the duration of the focus. The different colors in the hot spot map represent the first average fixation time when the information of the guidance mark is recognized. According to its size, it is represented by the following: red > yellow > green > gray—that is, red represents the longest first average fixation time, followed by yellow and green, and gray represents the shortest average fixation time. The number of visits in the circle represents the number of total visits, and the size of the circle represents the length of stay. Taking a guidance mark plate in path 3 as an example of the different observation levels of each subject, heat maps of three groups of changing factors were obtained after averaging, as shown in Table 2.

In Figure 6a—that is, the current scheme—the location of the mark is 17.05 m, the height of the mark is 3.0 m, the deflection angle of the mark is 53°, and the visual behavior comprehensive index is 0.2136, obtained through eye movement experiments. In Figure 6b, according to the adaptive aging safety design, the location of the mark is 15.81 m, the height of the mark is 2.73 m, and the deflection angle of the mark is 37°; the scheme was adjusted to obtain a comprehensive visual behavior index of 0.2391, which indicates that the adaptive aging safety design has a significant optimization effect on the visual cognition of the elderly population. In Figure 6c, considering the further improvement of the cognitive effect, the position of the mark is finely adjusted to 15.69 m, the height of the mark is adjusted to 2.70 m, the deflection angle of the mark is adjusted to 34°, and the visual behavior comprehensive index is 0.2421—the radar charts are shown in Figure 6.

According to Figure 6b,c, the calculation of the visual behavior comprehensive index improvement of two optimizations of guidance marks’ adaptive aging design of path 3 was as follows:$$\{\begin{array}{c} \frac{0.3470-0.3136}{0.3136}\times 100\%=10.65\%{\;1}_{th}\;\mathrm{optimization}\\ \frac{0.3577-0.3136}{0.3136}\times 100\%=12.90\%{\;2}_{th}\;\mathrm{optimization} \end{array}$$

According to the above analysis, the comprehensive visual behavior index of the first optimization of the adaptive aging design of the guidance marks of path 3 was increased by 10.65%, the comprehensive visual behavior index of the second optimization was increased by 12.90%, and the improvement of the second optimization was less than 2%, compared with the first optimization. The optimization results are satisfactory. Therefore, the second optimization scheme can be used for the adaptive aging design of the guidance marks of path 3. Then, in optimizing the other paths, the optimization results were as follows: path 1 was increased by 13.36%, path 2 was increased by 7.42%, path 4 was increased by 15.64%, path 5 was increased by 13.14%, and path 7 was increased by 3.53%. Based on case analysis, this design method can guide the adaptive aging design of guidance marks in rail transit connection systems.

## 5. Conclusions

This paper proposes an adaptive aging safety design for the guidance marks of rail transit connection systems, based on the elderly population′s eye movement data and visual recognition space system. The range of the elderly population′s visual recognition space is determined by the visual angle, the visual distance, and the height of the elderly population. Combined with the current situation of rail transit connection, the proposed preliminary adaptive aging design of the guidance marks is carried out by setting the indicators of position, height, and deflection angle. In the rail transit connection system, the first fixation time of 37 elderly people on a single mark, a flow line, and an area, the fixation time of a certain type of mark in a flow line, and the number of visits to a certain type of mark in the whole flow line, are collected, respectively. This was carried out following the principle that each person only walked each connection path once, ensuring that the subjects were not familiar with the environment. Through the analysis of eye movement data, it was concluded that the visual behavior composite indexes of the six connection path guidance marks that needed to be improved were improved by 11.35%, 7.42%, 13.34%, 15.64%, 13.14%, and 3.53%, respectively. The results prove the effectiveness and superiority of the modified adaptive aging safety design. This design effectively solves the extant problems, such as improper setting height, unreasonable setting position, and inappropriate deflection angle of the guidance marks of the rail transit connection system; this solution alleviates the visual cognitive demand on the elderly population when comprehending the guidance marks of rail transit connection systems. Considering the eye movement data, the visual behavior composite index is proposed from three aspects, as follows: fixation duration, first fixation duration, and number of visits. These aspects further optimize the design of the guidance marks and improve the degree of the adaptive aging design of the guidance marks. This research provides a theoretical basis and design method reference for improving the cognition of traffic mark information for the elderly population, implementing the renovation project of traffic adaptive aging facilities, and ensuring travel safety.

## Figures and Tables

**Figure 1 ijerph-19-00725-f001:**
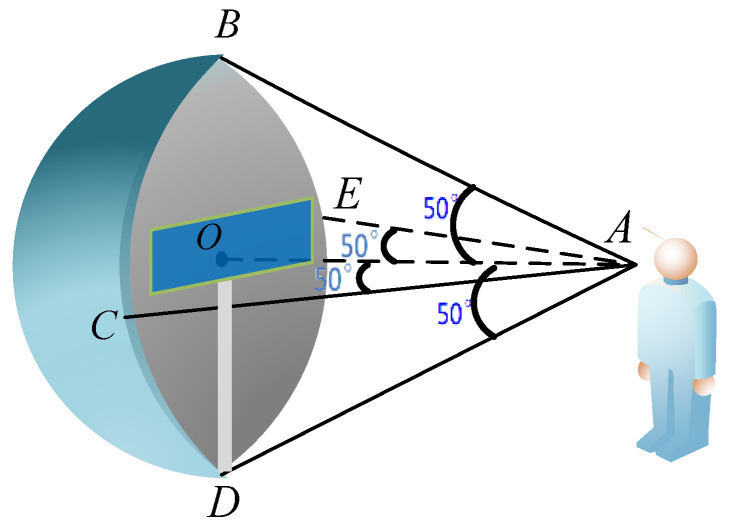
Visual field of the elderly population.

**Figure 2 ijerph-19-00725-f002:**
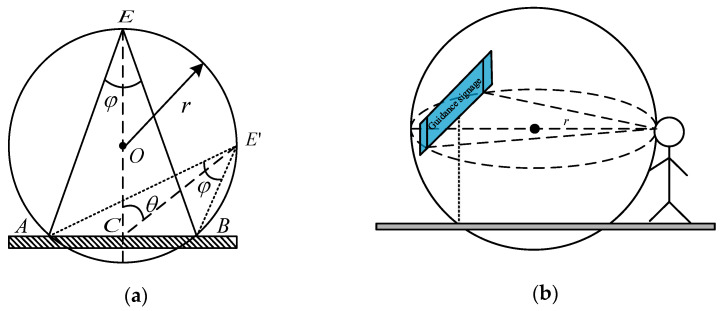
Visual range of mark. (**a**) Visual range model, (**b**) Visual range of mark.

**Figure 3 ijerph-19-00725-f003:**
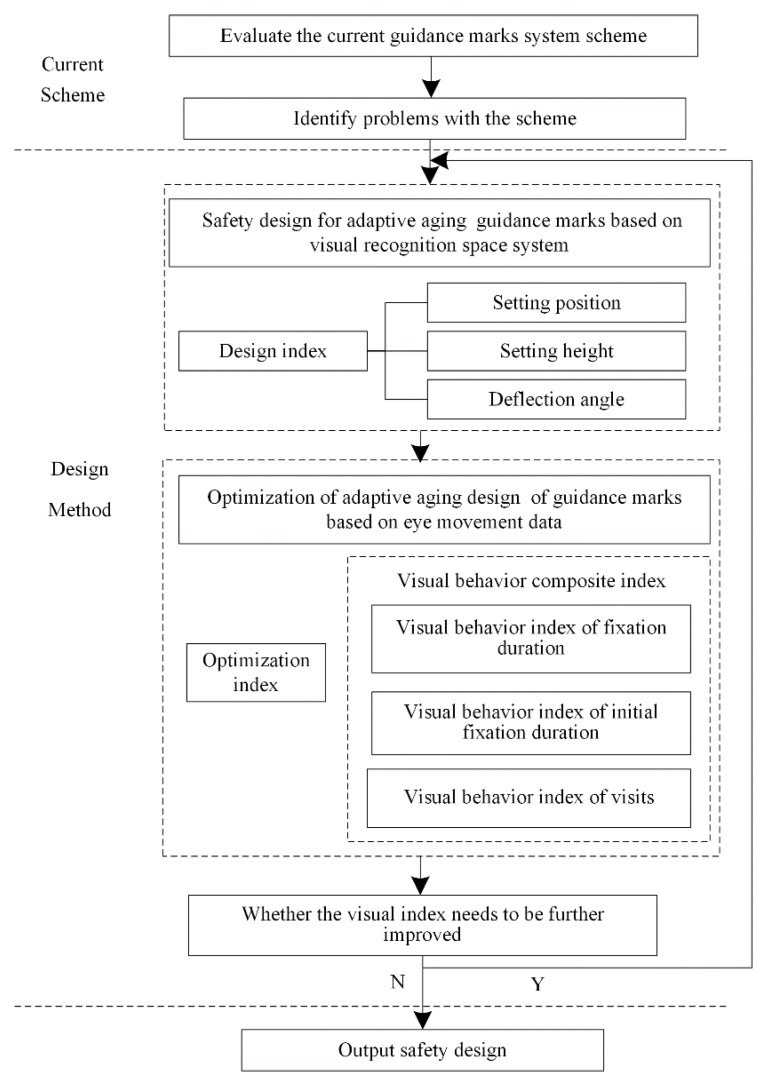
Safety design process of adaptive aging mark.

**Figure 4 ijerph-19-00725-f004:**
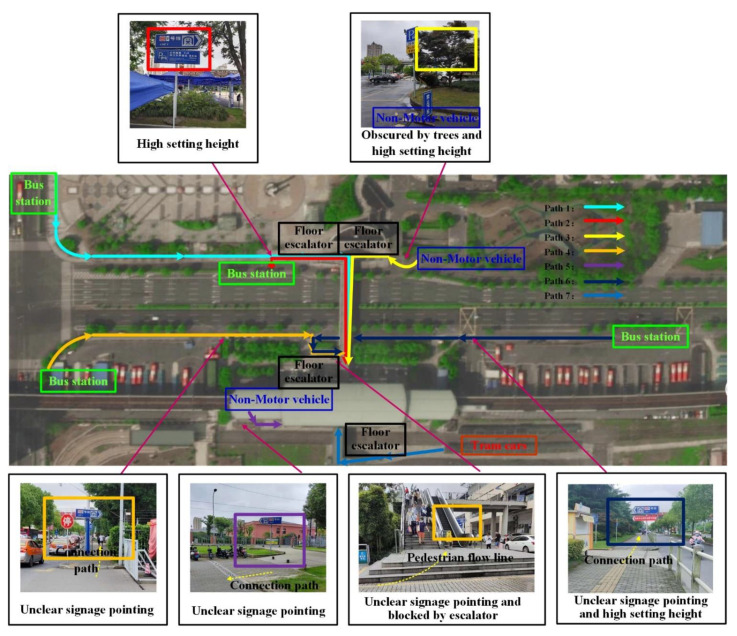
Songjiang University Town Subway Station.

**Figure 5 ijerph-19-00725-f005:**
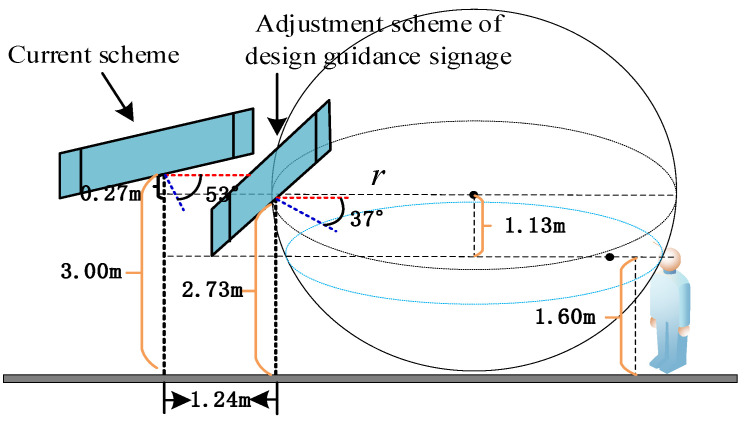
Guidance mark designed for adaptive aging.

**Figure 6 ijerph-19-00725-f006:**
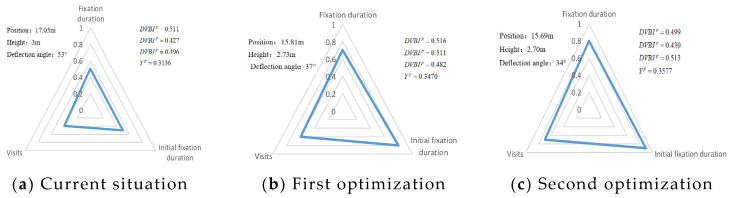
Radar diagram of path 3 guidance mark for adaptive aging optimization design. (**a**) Current situation; (**b**) First optimization; (**c**) Second optimization.

**Table 1 ijerph-19-00725-t001:** Eye movement data of the connection paths of Songjiang University Town Station.

Optimal Situation		Path					
		1	2	3	4	5	7
Current Situation	$DVBI^{p}$	0.515	0.451	0.511	0.516	0.618	0.625
	$FVBI^{p}$	0.503	0.523	0.427	0.500	0.478	0.609
	$NVBI^{p}$	0.497	0.412	0.496	0.520	0.510	0.590
	$Y^{p}$	0.2777	0.3650	0.3136	0.3632	0.4472	0.4472
First Optimization	$DVBI^{p}$	0.512	0.42	0.516	0.391	0.620	0.623
	$FVBI^{p}$	0.498	0.534	0.511	0.478	0.499	0.614
	$NVBI^{p}$	0.430	0.378	0.482	0.510	0.509	0.579
	$Y^{p}$	0.2812	0.3895	0.3470	0.3911	0.5049	0.4605
Second Optimization	$DVBI^{p}$	0.514	0.424	0.499	0.430	0.620	0.619
	$FVBI^{p}$	0.501	0.541	0.430	0.490	0.508	0.617
	$NVBI^{p}$	0.482	0.396	0.513	0.520	0.510	0.583
	$Y^{p}$	0.3148	0.3921	0.3577	0.4200	0.5060	0.4630

**Table 2 ijerph-19-00725-t002:** Eye movement data hotspot chart.

Eye Movement Date	Current Situation	First Optimization	Second Optimization
First Average Fixation Time	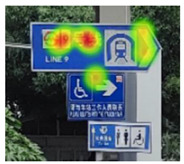	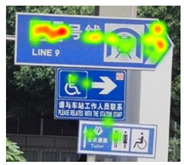	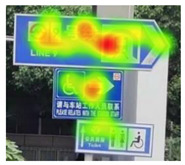
Visits	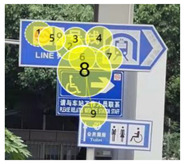	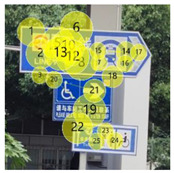	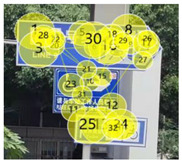

## Data Availability

The data used in this research were provided by National Natural Science Foundation of China at Shanghai University of Engineering Science. The data are available when readers ask the authors for academic purposes through email address fangyong87@tongji.edu.cn.

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
