# Peer review of "Adaptive Aging Safety of Guidance Marks in Rail Transit Connection Systems Based on Eye Movement Data"

_ijerph, 2022, doi:10.3390/ijerph19020725_

Round 1

Reviewer 1 Report

Suggestion for authors or for any other further research refering to design index shape type could be implemented as additional parameter to be considered as visual recognition space system;

Author Response

Point : Suggestion for authors or for any other further research refering to design index shape type could be implemented as additional parameter to be considered as visual recognition space system.

Response: Thank you very much for your advice.In the methods part 3.1, we explained the visual recognition space system and added Step 4 Mark layout design.

Page 3, lines 116-125: Based on the visual characteristics of the elderly and mark recognition logic, this paper constructs a visual recognition space system for the elderly, which can be understood as a three-dimensional space within which the human eye can capture the mark and clearly identify the information on the mark. Therefore, the visual recognition space system is composed of three parts: the visual space, the visual range of the mark, and the visual recognition information of the mark. Determine the visual space of the elderly from the three aspects of visual angle, visual distance and height of the elderly, from the deflection angle, height, position of the mark to deter-mine the visual range of the mark and design parameters such as the layout elements of the mark itself identify the visual recognition information, so as to construct the visual recognition space system of the elderly.

Page 5, lines 179-183: Step 4: Mark layout design.

The mark layout elements are designed in accordance with the corresponding specifications, including the type, shape, size, color, graphic and text. This paper uses eye movement indicators to indirectly characterize the impact of the mark information on the visual recognition of the elderly.

Reviewer 2 Report

This paper presents an interesting and very significant area of work with important implications for societal safety and health. The article is structured well, and the sections flow coherently. The presented research overview provides a fitting review of relevant related works, and sets the context for the study. The research methods design is appropriate for the study, and the results and findings are adequately described.

Some minor comments to improve the manuscript are as follows: 
I recommend the authors provide a simple explanation of the "adaptive aging safety design" concept in the introduction. 
I recommend proof-reading and editing the entire manuscript to fix minor spelling errors
e.g. 3.1 Annlysis of visual characteristics
line 73 - Diretion sense
line 104 - passenders

Author Response

Point 1: I recommend the authors provide a simple explanation of the "adaptive aging safety design" concept in the introduction.

Response 1: Thank you very much for your advice.In the introduction, we explained adaptive aging safety design. It is a design concept that based on the physiological characteristics of the elderly and can meet the safety of the elderly.

Point 2: I recommend proof-reading and editing the entire manuscript to fix minor spelling errors.

e.g. 3.1 Annlysis of visual characteristics

line 73 - Diretion sense

line 104 - passenders

Response 1: Thank you very much for your advice. Upon inspection, we found that the entire manuscript still contained some typos and grammatical errors and they have been corrected(line 35、39、43、49、75、90、106、287、295).

Reviewer 3 Report

This research article presents an important investigation based on eye movement data. The manuscript, based on the visual characteristics of the elderly. First the system constructs the visual recognition space of the elderly and then an adaptive aging safety design method for the guidance marks in the rail transit connection system is proposed. The research is important in this field (considering that focus on a special group, the elderly). This reviewer thinks that this manuscript can be publish with some minor corrections. This reviewer encourages authors to take care of the following.

Please review the written English. There are several misspellings in the paper. For example, lines 39, 49,

Review the expressions (1), (2), (3) and (4), expression 1 corresponds to line BE and expression 2 corresponds to line AE, please check the expressions.

Figure 4 please change red colour by other more readable for a better understanding of the case study.

Within section 4.1.2 please justify the decision made referred to the subjects, “follow the principle that each connection route should only be taken once per person”

In page 10 of 12 please explain the colours of the fixation hot spot map

In figure 6 what happen with (a) and (c)? Please correct.

Please state in conclusions the utility of this investigation to show the practical and important use of the research.

Author Response

Point 1: Please review the written English. There are several misspellings in the paper. For example, lines 39, 49.

Response 1: Thank you very much for your advice. Upon inspection, we found that the entire manuscript still contained some typos and grammatical errors and they have been corrected(lines 35、39、43、49、75、90、106、287、295).

Point 2: Review the expressions (1), (2), (3) and (4), expression 1 corresponds to line BE and expression 2 corresponds to line AE, please check the expressions.

Response 2: Thank you very much for your advice. we revised the expressions.

Point 3: Figure 4 please change red colour by other more readable for a better understanding of the case study.

Response 3: Thank you very much for your advice. we revised Figure 4 again .

Point 4: Within section 4.1.2 please justify the decision made referred to the subjects, “follow the principle that each connection route should only be taken once per person”

Response 4: In order to avoid the impact of repeated experiments on the objectivity of the data,and follow the principle that each connection route should only be taken once per person.(lines 296、297)

Point 5: In page 10 of 12 please explain the colours of the fixation hot spot map

Response 5: The different colors in the hot spot map represent the first average fixation time when the information of the guidance mark is recognized. According to its size, it is repre-sented by: red > yellow > green > gray, that is, red represents the longest first average fixation time, followed by yellow and green, and gray represents the shortest average fixation time.(lines 307-311)

Point 6: In figure 6 what happen with (a) and (c)? Please correct.

Response 6: In the figure 6(a), the location of the mark is 17.05m, the height of the mark is 3.0m, the deflection angle of the mark is 53°, and the visual behavior comprehensive index is 0.2136 obtained through eye movement experiments. In the figure 6(b), according to the adap-tive aging safety design, the location of the mark is 15.81m, the height of the mark is 2.73m, and the deflection angle of the mark is 37° and the scheme is adjusted to obtain a comprehensive visual behavior index of 0.2391., which indicates that the adaptive aging safety design has a significant optimization effect on the visual cognition of the elderly. In the figure 6(c),considering the further improvement of the cognitive effect, the position of the mark is finely adjusted to 15.69 m, the height of the mark is 2.70m, the deflection angle of the mark is 34°, the visual behavior comprehensive index is 0.2421, and the radar chart is drawn as shown in Figure 6.(lines 318-327)

Point 7: Please state in conclusions the utility of this investigation to show the practical and important use of the research.

Response 7: Thank you very much for your advice.This research provides a theoretical basis and design method reference for improving the cognition of traffic mark information for the elderly, implementing the renovation project of traffic adaptive aging facilities, and ensuring travel safety.(lines 368-371)
